# A radiative-convective model computing precipitations with the maximum entropy production hypothesis.

Quentin Pikeroen[1], Didier Paillard[1], and Karine Watrin[1]

[1]Université Paris-Saclay, CNRS, CEA, UVSQ, Laboratoire des sciences du climat et de l'environnement, 91191, Gif-sur-Yvette, France.

**Correspondence:** Pikeroen (quentin.pikeroen@lsce.ipsl.fr), Paillard (didier.paillard@lsce.ipsl.fr), Watrin (karine.watrin@protonmail.com)

**Abstract.**

All climate models use parameterisations and tuning in order to be accurate. The different parameterisations and tuning processes are the primary source of difference between models. Because models are tuned with present observations of Earth, they may not simulate accurately climates of other planets or paleoclimates. A model with no adjustable parameter that happens to fit today's observations is probably more universal and should be more appropriate to model paleoclimates. However, to our knowledge, such a model does not exist or is yet to be developed. This paper aims to improve a parameter-free radiative-convective model that computes a realistic temperature vertical profile to compute the water cycle, giving a value on average tropical precipitations. Although it is known that the radiative transfer constrains the order of magnitude of precipitation, no parameter-free model has yet been able to compute precipitation. Our model finds a precipitation value closer to observations than similar radiative-convective models or some GCMs.

## 1 Introduction

Historically, climate models have evolved from elementary conceptual models to energy balance models (EBMs), then to radiative-convective models (RCMs), and after that, General Circulation Models (GCMs), and finally, the state-of-the-art Earth System Models (ESMs) (Paul N Edwards, 2011); with a constant increase in complexity and calculation rate.

Researchers generally consider GCMs and ESMs "the best" models because they account for many phenomena. Indeed, if some specific part of the model does not fit well enough with observations, it is always possible to spend time adding more complexity and making it fit better. There is a hope that if we put enough work into it, GCMs or ESMs will be very close to observations. Furthermore, these models cover the entire earth, accurately describing the position of oceans and continents, the orography, the cryosphere, and the vegetation with a resolution now below a hundred kilometres. Thus, they are beneficial to answer specific questions, like taking an example out of many, how crop yield would evolve in a particular area within this century. Today's ESMs predict very robust temperature changes for increasing levels of $CO_2$ (see AR6 IPCC Fig. 4.19) for most regions of the globe. On the contrary, they do not predict robust changes in precipitations (AR6 IPCC Fig. 4.24). Individual models show opposite signs of precipitation changes in some regions (fig 4.42d of AR6 IPCC). Even when looking for global

mean changes when increasing $CO_2$, temperatures show much less uncertainty than precipitations (AR6 IPCC fig4.2a-b). The reason for such uncertainty lies in the difficulty of parameterising the equations for water fluxes.

Indeed, the atmospheric part of GCMs or ESMs is based on the Navier-Stokes equations, whose length scales range from $L = 10^3$ km to less than $\eta = 10$ mm, the viscous or the Kolmogorov scale of the atmosphere. The number of modes required to model every scale is $N = (L/\eta)^3 \approx 10^{24}$, by far unreachable by today's computer (we would need $10^8$ times the full storage capacity of today's supercomputer to store one time-step). To deal with this problem, climate computers integrate only large scales with the Navier-Stokes equation, and sub-grid processes are parameterised differently for every model, leading to different results. Although, the physics of every model is the same. How much the parameterisations affect our ability to truly predict climate is a deep and open question (Hourdin et al., 2017; Dommenget and Rezny, 2018). Today's ESMs have hundreds of adjustable parameters.

Looking at global mean evaporation in the tropics, GCMs from the atmospheric model intercomparison project have a mean positive bias of 20 $W.m^{-2}$ (Zhou et al. (2020)). Radiative-convective models compute similar values of mean evaporation to GCMs (Betts and Ridgway (1988); Rennó et al. (1994); Takahashi (2009)). While radiative-convective models approximate spatial mean quantities well, they are sometimes close to observations at finer scales (Jakob et al. (2019)). O'Gorman and Schneider (2008) showed that a GCM gives similar output to a radiative-convective model (using the same simple radiative code and adequate parameterisation of surface wind speed for the radiative-convective model), for example, a similar relationship between temperature and precipitation is found (figure 3 and 4 of O'Gorman and Schneider (2008)). Eq. 19 of Takahashi (2009) gives an upper bound for the surface latent heat flux (so to evaporation, equal to precipitation in a stationary setting). The exact evaporation value depends on the sensible heat flux and surface radiative transfer. However, if the same drag coefficient and wind speed are used to compute surface sensible and latent heat flux, then because of the exponential dependence of mixing ratio with temperature, latent heat flux dominates sensible heat flux (see eq. 7 and 8 of Rennó et al. (1994)). Thus, the only knowledge of the surface radiative transfer gives an upper limit to the evaporation. Therefore, in our goal to compute precipitation with a parameter-free model, it is relevant to use a radiative-convective model.

The efficiency of radiative-convective models in reproducing accurate surface temperature or climate sensitivity comes from 3 ingredients (Jeevanjee et al. (2022)): 1) a realistic computation of $CO_2$, $O_3$ and water vapour radiative transfer, 2) the convective adjustment, where all unstable lapse rates are adjusted to the moist adiabatic lapse rate of -6.5 $K.m^{-1}$, 3) a fixed relative humidity in the radiative code to account for the water vapour feedback. Manabe and Strickler (1964) introduced the convective adjustment, and Manabe and Wetherald (1967) the fixed relative humidity. Cloud microphysics was already present in the Manabe and Strickler (1964) model. Sarachik (1978) added a flux between the atmosphere and the ocean and found that in a model with no clouds and a surface with no heat capacity, the absence of the ocean compensates for the neglecting of clouds. Rennó et al. (1994) computed explicitly the moisture profile with a cumulus convection scheme, leading to a wider model sensitivity to parameter changes. Recent developments in radiative-convective modelling focus on improving the cloud parameterisation, creating a 3D radiative-convective model, or using it in a GCM (Wing et al. (2018)). Radiation-convective models mainly follow the same tendency as GCM and ESM models. More and more complexity is added, and more and more parameters are present. All models are tuned with observations. Because of this tuning, we do not know whether these

models are accurate in climates very far from the present Earth condition, such as paleoclimates. The first "parameter" is the convection scheme in a radiative-convective model with a convective adjustment: the algorithm by which which the lapse rate adjusts to the moist adiabatic can change the results. Also, a small diffusion of temperature and moisture is used to stabilise the numerical scheme. Most importantly, the surface drag coefficient and wind speed are prescribed to values unknown in paleoclimates, though they impact the surface energy budget. Indeed, the sensible and latent heat flux is usually calculated from gradients using a Fickian law, with a coefficient adjusted on observations. Since the physical (conservation) laws are the same for all climate models, the main differences between them are mostly linked to the tuning of these parameters: differences are therefore linked to different model settings, different choices of parametric formula, different tuning strategies, or different tuning data sets. It is probably one of the main problems in climate modelling.

To overcome this issue, we build a radiative-convective model with zero adjustable parameters in this study. Because the method used is completely new and has been little investigated, the model is built from scratch and appears to be a jump back into the 70s. Many issues could be raised: the model is cloud-free (and we do not know how to compute clouds without parameters), has a ground with no heat capacity, an infinite water reservoir, relative humidity is fixed to climatology in the radiative code and to 1 in the energy fluxes computation, albedo and solar constant are fixed, stationarity is assumed, and the model is 1D radiative-convective. However, the article's purpose is not to build a code as accurate as today's complex and highly tuned codes but to build a model as accurate as equivalent radiative-convective models built in the 70s and 80s. If it is possible to obtain similar or better results with a parameter-free model than a similar but tuned model, it would pave the way for creating a full climate model, as has been done for tuned models since the 70s, but this time not using tuning.

To get rid of parameters, the unknown variables are determined with a variational problem, the maximum of entropy production (MEP). The idea is to express the variational problem (entropy production and constraints) as a function of the unknown variables, like energy fluxes or water vapour fluxes, so they will adjust themselves to maximise entropy production. Therefore, we do not parameterise them. One could argue that MEP is just another way to parameterise. However, it is very different from the usual data assimilation techniques (see, for example, Lopez (2007) for precipitation and clouds) because it does not use any observational data about the variable of interest to predict it (like temperatures or precipitation). A MEP model was first used for climate by Paltridge (1975) to predict meridional fluxes and showed good agreement with observations. Unfortunately, it also had some parameterisations.

The MEP is only a hypothesis and lacks rigorous mathematical proof. However, it seems very general and is used in many domains, like crystal growth, electric charge transfer, biological evolution, and many others (Martyushev and Seleznev, 2006). First, the MEP hypothesis must not be mistaken with the second law of thermodynamics, which only states that entropy production is positive ($\sigma \geq 0$) but not necessarily maximised. Next, when equilibrium thermodynamics and entropy are reinterpreted with the formalism of information theory, it is possible to obtain the main results of equilibrium thermodynamics (Jaynes, 1957) using the maximum entropy principle (MaxEnt). A possible way to understand the maximum entropy production hypothesis is to see it as the non-equilibrium or time derivative "equivalent" of MaxEnt. Following this idea, Dewar (2003) and Dewar (2005) tried to prove the MEP hypothesis using MaxEnt. However, to quote Martyushev (2021), "Dewar's argument not only involves a number of nonobvious fundamental assumptions but also is nonrigorous and erroneous in a number of points.". The

MEP hypothesis has also been related to other variational approaches in fluid dynamics or climate (Ozawa et al., 2003), like
the *maximum dissipation rate* when temperatures are fixed (Malkus, 1956), or the *maximum generation of available potential
energy* when in a steady-state (Lorenz, 1960). We do not try to demonstrate the MEP hypothesis here, but we prefer modelling
the climate and the water cycle using it. Suppose results happen to be close to observations. In that case, we let as an open
problem for theoreticians the explanation of why it works in our particular case (see Martyushev (2021) for a recent general
review on MEP hypothesis).

In climate, it has been used to predict oceanic or atmospheric horizontal heat fluxes (Grassl, 1981; Gerard et al., 1990;
Lorenz et al., 2001; Paltridge et al., 2007; Herbert et al., 2011), as well as vertical heat fluxes (Ozawa and Ohmura, 1997; Pujol
and Fort, 2002; Wang et al., 2008; Herbert et al., 2013b), or both horizontal and vertical (Pascale et al., 2012); where always
the entropy production of heat transfers due to atmospheric turbulence is maximised under some constraints. A limitation of
using the MEP hypothesis (and variational formulations in general) is that all the variables of the variational problem need to
be solved at once, making it difficult to add new phenomena. Another limitation is that to get meaningful results, we have to put
on the variational problem constraints that are physically relevant and represent the main processes of the atmosphere, (Goody,
2007; Dewar, 2009). It is far to be obvious and only sometimes easily solvable. Moreover, as the driver for heat transfers is the
input of radiative energy in the system, the radiative code must be accurate to have results close to observations. Otherwise, we
are limited to qualitative (Lorenz et al., 2001) but not quantitative (Goody, 2007) agreement with data. In previous studies, MEP
has generally been used for straightforward cases or with additional parameters. For 1D-vertical models, a grey atmosphere was
used, and gravity was not considered. Recently, a new radiative-convective model was created with a more realistic radiative
code (Herbert et al., 2013b). Moreover, geopotential and latent heat were added (Labarre et al., 2019). This model already
produces reasonable temperatures. This model is not based on a convective adjustment like Manabe and Wetherald (1967), but
has the same crucial physical ingredients, allowing it to give similar results. However, the dynamical part (i.e., non-radiative)
is treated without any adjustable parameter, giving the MEP model more universality because no one has ever told the model
which values to take in any standard configuration. If the results are close to observations, it alerts us that there might be
something powerful in the MEP hypothesis or the imposed constraints. In this study, the same model was used, and a new
constraint on the water cycle was added, leading to a prediction of precipitations. The fact that this procedure can produce
suitable water vapour fluxes has never been shown before.

## 2 The radiative code

The radiative code used here is the one of Herbert et al. (2013b) (see their supplemental material for details) and is based on
the Net-Exchange formulation (Dufresne et al. (2005)). It is more advanced than the grey atmosphere models used in previous
studies (Ozawa and Ohmura, 1997), thus leading to results closer to observations. Let us consider an atmosphere divided into

$n+1$ layers on the vertical axis (see Fig. 1). The net radiative energy budget $\mathcal{R}_i$ (i.e. the input of energy thanks to radiation) writes :

$$\mathcal{R}_i(T, q, O_3, CO_2, \alpha) = SW_i(q, O_3, \alpha) + LW_i(T, q, CO_2) \tag{1}$$

where $SW_i$ and $LW_i$ are the solar and infrared net energy budgets. $q = M_{water}/M_{air}$, $O_3$ and $CO_2$ are prescribed vertical profiles of specific humidity, ozone and carbon dioxide concentrations, given by page 3 of McClatchey (1972), corresponding to a standard atmosphere, based on averaging observations. To take into account the water vapour feedback with temperature, the relative humidity profile $h = q/q_s(T)$ is fixed for the computation of $\mathcal{R}_i$, so that $q$ varies with $T$. $\alpha$ is the albedo of the surface. $T$ is the temperature profile. The parameters $h$, $O_3$, $CO_2$, and $\alpha$ are fixed, so the energy budget $\mathcal{R}_i$ is a function of the temperatures only. Note that $T = \{T_j\}_{j=0,\ldots,n}$, where $T_j$ is the temperature in box $j$, hence $\mathcal{R}_i(T)$ is a functional of the temperatures, i.e. a non-local function of T. That will be important for the variational problem.

Usually, when computing the radiative energy budget, one uses a local description of the radiative energy fluxes. In the NEF framework (Dufresne et al., 2005), the description of energy transfer is global. Each radiative energy input in the layer $i$ is broken down into the individual contributions of all atmospheric layers. The net energy exchange rate between layer $i$ and $j$ is $\psi_{ij}$. At this point, this formulation is strictly equivalent to the usual one. However, it makes it easier to develop approximations that reduce computational time while automatically satisfying the fundamental laws of physics: The energy exchange rate is antisymmetric ($\psi_{ij} = -\psi_{ji}$). Thus, the total energy is conserved ($\sum_{i,j}\psi_{ij} = 0$) and also the second law of thermodynamics is satisfied by keeping $\psi_{ij}$ the same sign as $T_j - T_i$. Yet, because the resolution of the variational problem is strongly sensitive to the constraints imposed, it is of utmost importance that the laws of physics are rigorously satisfied.

To approximate the radiative transfer, the infrared spectrum is divided into 22 narrow bands, and the absorption coefficient for water vapour and carbon dioxide is calculated with the Goody (1952) statistical model and the data from Rodgers and Walshaw (1966). The diffusive approximation is made with the diffusion factor $\mu = 1.66$ for spatial integration. In the visible domain, water vapour and ozone absorption are computed with the parameterisation of Lacis and Hansen (1974).

## 3 The maximisation of entropy production

### 3.1 Previous setting: Energy conservation

This constraint has been used alone in Herbert et al. (2013b). The atmosphere is divided into $n+1$ layers on the vertical axis. Each layer $i$ has a temperature $T_i$, a variable of the variational problem. Between layer $i$ and $i+1$, there is a non-radiative energy flux $F_i$, whose nature is not explicit yet (with no additional constraints, it may be interpreted as conduction). When considering a steady state, the total energy balance reads :

$$\mathcal{R}_i = F_{i+1} - F_i \tag{2}$$

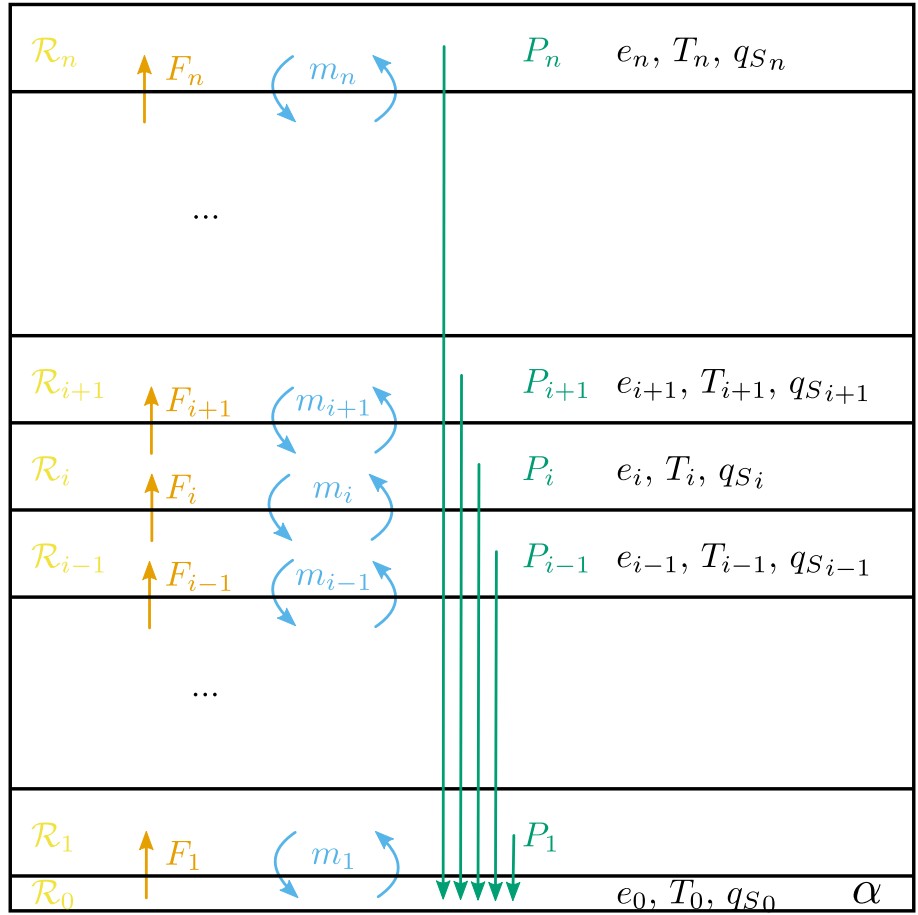

**Figure 1.** Scheme of the 1D vertical box model with the box 0 being infinitely small, and other boxes are separated by equal level of pressure. $\mathcal{R}_i$ is the radiative energy balance, and depends on all temperatures. $F_i$ is the energy flux between adjacent boxes. $m_i$ is the air mass flux. $e_i$ is the specific energy (and depends on the local temperature), $T_i$ is the temperature, and $q_{S_i}$ is the relative humidity taken at saturation (depends on the local temperature).

Where $F_0$ and $F_{n+1}$ are taken equal to zero as if no energy (other than radiative) goes to the space or comes from the ground. The entropy production associated with these energy fluxes writes:

$$155 \quad \sigma = -\sum_{i=1}^{n} F_i \left( \frac{1}{T_{i-1}} - \frac{1}{T_i} \right) \overset{(2)}{=} \sum_{i=0}^{n} -\frac{\mathcal{R}_i(T)}{T_i} \tag{3}$$

At stationary state, the total input of radiative energy must be equal to zero:

$$\sum_{i=0}^{n} \mathcal{R}_i(T) = 0 \tag{4}$$

The entropy production (eq. 3) is maximised under the constraint of energy conservation (eq. 4), leading to the following variational problem:

$$\max_{(T_0,\dots,T_n)} \left\{ \sum_{i=0}^{n} -\frac{\mathcal{R}_i(T)}{T_i} \,\middle|\, \sum_{i=0}^{n} \mathcal{R}_i(T) = 0 \right\} \tag{ENERGY}$$

It is the exact same problem as eq. 24 of Herbert et al. (2013b). The predicted lapse rate has then the correct magnitude in the lower troposphere but not in the upper layers (see blue square points Fig. 2a).

### 3.2 Previous setting: Air convection

Air convection has been studied in Labarre et al. (2019). So far in the article, the atmosphere's internal energy or air mass fluxes have not been expressed. It might give more physical results to put in the variational problem a constraint on how air masses transport the energy in the atmosphere. By taking into account the sensible heat, the gravitational energy, and the latent heat, the specific energy (the energy per unit mass) writes:

$$e_i = C_p T_i + g z_i + L q_i \tag{5}$$

where $C_p$ is the heat capacity of the air, $g$ the standard acceleration due to gravity, $z_i$ the elevation of layer $i$, $L$ the latent heat of vaporization, and $q_i$ the specific humidity of water vapour ($m_{water}/m_{air}$). The elevation $z_i$ is expressed as a function of the temperatures below (see appendix A of Labarre et al. (2019)), and the specific humidity $q_i$ is taken equal to its value at saturation, $q_i = q_s(T)$. Consequently, the specific energy $e_i(T)$ is like $\mathcal{R}_i(T)$ a functional of temperatures.

Now, take a mass flow rate $m_i$ between layer $i-1$ and $i$. Suppose the air is transported adiabatically and then thermalizes once in the layer $i$. It means that the energy in the air mass taken from layer $i-1$ is entirely transported to layer $i$. Thus, the upward energy transported equals $m_i e_{i-1}$. To easily conserve the air in all boxes, the same amount of air $m_i$ is adiabatically taken from layer $i$ to layer $i-1$, transporting downward energy equal to $m_i e_i$. Thereby, the total energy flux between layer $i-1$ and $i$ writes :

$$F_i = m_i(e_{i-1} - e_i) \tag{6}$$

Of course, for the reasoning to be consistent, we must have $m_i \geq 0$, which gives a new constraint on the energy fluxes and the temperatures. The following equation summarizes the variational problem:

$$\max_{(T_0,\dots,T_n),(m_1,\dots,m_n)} \left\{ \begin{array}{l} \sum_{i=0}^{n} -\frac{\mathcal{R}_i(T)}{T_i} \,|\, \sum_{i=0}^{n} \mathcal{R}_i(T) = 0 \text{ and } m_i \geq 0 \\[2ex] \text{with } \mathcal{R}_i = F_{i+1} - F_i,\, F_i = m_i(e_{i-1}(T) - e_i(T)) \end{array} \right\} \tag{CONV}$$

It is the same variational problem as eq. 11 of Labarre et al. (2019) (although expressed as a function of temperature instead of energy fluxes). The additional constraint allows for a much better simulation of the upper atmosphere (see orange diamonds Fig. 2).

## 3.3 New setting: Water transport and precipitation

The model's novelty described here comes from the addition of water transport and precipitation. In the formulation above, water vapour is a function of the temperature only and thus has no reason to be conserved when transported by the air masses. Infinite levels of water vapour could be created or disappear. In this study, we add a constraint on conserving water vapour that is supposed to mimic precipitations. We impose that water vapour cannot appear when transported, but it can disappear, and we call this phenomenon precipitation as if water vapour was transformed into liquid water. The flow rate $m_i$ transports upward between layer $i-1$ and $i$ an amount of water equal to $m_i q_{i-1}$, where $q_i$ is the specific humidity of water vapour in the air, and it transports downward an amount of water equal to $m_i q_i$. Then the water flux between layers $i-1$ and $i$ writes (similarly to eq. 6):

$$F_i^w = m_i(q_{i-1} - q_i) \tag{7}$$

The amount of water vapour that disappears in layer $i$ is written :

$$P_i = F_{i+1}^w - F_i^w \tag{8}$$

Where $F_0^w$ and $F_{n+1}^w$ are taken equal to zero as if no water comes from underground or goes to space. The layer $i=0$ is a surface boundary layer, supposed to be very thin, and plays the role of the surface. For $i=1,...,n$, we impose that $P_i \geq 0$, and $P_i$ is called precipitation. On the layer $i=0$, because $\sum_i P_i = 0$ we have:

$$P_0 = -\sum_{i=1}^{n} P_i \tag{9}$$

where $-P_0$ is the evaporation in layer $i=0$ and is equal to the total precipitations. The specific humidity $q_i$ is considered equal to its value at saturation $q_s(T)$ and then depends only on the temperature. The variational problem can be summarized by:

$$\max_{(T_0,...,T_n),(m_1,...,m_n)} \left\{ \begin{array}{l} \sum_{i=0}^{n} -\frac{\mathcal{R}_i(T)}{T_i} \mid \sum_{i=0}^{n} \mathcal{R}_i(T) = 0 \text{ and } m_i \geq 0,\, P_i \geq 0 \text{ for } i \geq 1 \\ \\ \text{with } \mathcal{R}_i = F_{i+1} - F_i,\, F_i = m_i(e_{i-1}(T) - e_i(T)), \\ P_i = F_{i+1}^w - F_i^w,\, F_i^w = m_i(q_{S_{i-1}}(T) - q_{S_i}(T)) \end{array} \right\} \tag{PRECIP}$$

Although it looks like a heavy equation, it is, in fact, very short when saying absolutely all the physics of the model is contained in it.

## 4 Numerical resolution

The variational problems ENERGY, CONV and PRECIP are solved using a sequential quadratic programming algorithm (Kraft, 1988; Virtanen et al., 2020). The basic principle is that it takes some initial conditions and performs a gradient descent until it finds a local maximum. Such an algorithm mathematically converges to a global maximum for a convex problem;

however, for a non-convex problem, there is no guarantee that the maximum found is global. Several techniques that are not detailed here are used to get better results. One possibility is to manually test different initial conditions and see which gives the highest entropy production. For a given problem, every result presented here is the one with the highest entropy production found. Given that we tested many different initial conditions and found only a few local maxima, we are confident our results represent a global maximum. Nevertheless, it is still possible that another better result mathematically exists.

## 5 Results

We solve the problems with 21 boxes (1 surface boundary layer with albedo $\alpha$ and 20 atmospheric boxes), see Fig. 1, with prescribed standard vertical profiles of $O_3$, $CO_2$ concentrations and relative humidity taken from McClatchey (1972), corresponding to a tropical atmosphere. Indeed, because it is a 1D vertical model, taking midlatitude or boreal profiles would not make much sense, as horizontal fluxes become significant there. In the radiative code, the relative humidity $h = q/q_S(T)$ is fixed so that more $H_2O$ is present when temperature increases, which is an important positive feedback for climate sensitivity. However, in the variational problem, $q_i$ is still equal to $q_s(T)$ (eq. 5). This decoupling may be inconsistent, but it is common in the climate community or GCMs to examine models decoupled before studying fully coupled model configurations. We also choose to proceed in this way to keep the problem simpler and to be able to interpret results more easily. As in Rennó et al. (1994), incoming solar radiation is fixed to 342 W.m$^{-2}$, and albedo is equal to $\alpha = 0.1$.

Results for equations ENERGY, CONV and PRECIP are shown figure 2. For temperature and energy, the lowest point corresponds to the surface. For comparison, standard temperatures for a tropical atmosphere (based on observations) are plotted in figure 2a, taken from McClatchey (1972), as well as mean temperatures from the IPSL-CM6A-LR model between 23°S and 23°N. The observations are not taken in a specific area of the tropics. However, if we knew a place matching the requirements of the model, that is to say, where horizontal fluxes are negligible, and where we know the greenhouse gas concentration, it could be used and compared to observations at this place. The comparison is made here for spatially average tropical conditions, using standard greenhouse gas concentrations and temperatures, as is customary for radiative-convective models. Also, we only consider a stationary atmosphere, so the results should be interpreted only in a climatological context, not a meteorological one. The comparison should remain qualitative, as our model's purpose is not to fit the observations precisely but to give a "proof of concept" that obtaining relevant results with the MEP hypothesis is possible. First, when putting more and more constraints into the variational problem, it is expected that the entropy production found decreases because the space of possibilities wanes. We get this: the entropy production is equal to $\sigma = 53.9$ mW.m$^{-2}$.K$^{-1}$ for eq. ENERGY where only energy is conserved, $\sigma = 44.3$ mW.m$^{-2}$.K$^{-1}$ for eq. CONV where a specific pattern of mass fluxes transports energy and $\sigma = 41.1$ mW.m$^{-2}$.K$^{-1}$ for eq. PRECIP where water vapour is not allowed to appear.

### 5.1 With only energy conservation (blue square points)

Using only energy conservation has already been studied in Herbert et al. (2013b). Although the numerical resolution has been improved, the results here are very similar to Herbert et al. (2013b), but we show them for comparison with the next ones. The

only variables present in eq. ENERGY are temperatures $T$ and energy fluxes $F$. However, it is possible to compute afterwards variables like energy $e$ (5), mass fluxes $m$ (6), or precipitations $P$ (8), but of course, there is no reason for $m$ or $P$ to obey the constraint of positivity, and they do not. When looking at the entropy production equation (3), one sees two terms: an energy flux and a gradient of inverse temperature. To maximise the entropy production, variables find a balance between a state where energy fluxes are very high but temperatures homogeneous (so $\sigma = 0$), and a state where temperature gradients are high but energy fluxes are equal to zero (also $\sigma = 0$). Between the two, a balance is found where entropy production is maximised. Results are plotted in figure 2a for temperature and figure 2b for energy fluxes. As expected, temperatures are higher near the ground because the atmosphere is essentially heated by the solar radiation intercepted at $z = 0$. The energy fluxes compensate a bit for the temperature difference by going upwards (i.e. are positive Fig. 2(b)). The specific energy (eq. 5, Fig. 2c) decreases at the bottom because sensible heat $C_p T$ is the dominant term and increases at the top because gravity $gz$ becomes the dominant term. "Mass fluxes" are computed with eq. 6. Because $F$ is always positive, the sign of $m$ is always opposite to that of the energy gradients $\nabla e$. When these gradients are negative, $m$ is positive, but higher in the atmosphere when energy gradients become positive, the computed $m$ is then negative, which has no physical meaning regarding eq. 6. Then, imposing $m > 0$ becomes natural, as done in the following paragraphs.

## 5.2   With a pattern of convection (orange diamond points)

Using energy and mass conservation has already been studied in Labarre et al. (2019). Because the numerical resolution has been improved, results here differ slightly from Labarre et al. (2019). Adding the constraint of convection (6) forces the energy fluxes (Fig. 2b) to be opposed to the energy gradients. Thus, energy (Fig. 2c) gradients must remain negative to keep positive upward energy fluxes (energy must decrease with height). To do so, temperatures (Fig. 2a) adapt themselves to counteract the gravity, leading to a zone in the middle of the atmosphere where energy gradients equal zero and mass fluxes (Fig. 2d) equal $+\infty$; the atmosphere is perfectly mixed in this area, energy follows the moist adiabatic lapse rate (Fig. 2c, a vertical line). Thus, precipitations (Fig. 2e) are infinitely positive or negative. Higher in the sky, geopotential becomes too strong, energy gradients become positive and energy fluxes equal 0. This region defines the stratosphere at about $P \approx 250$ hPa and $z \approx 9$ km, where no convection occurs. By adding a convection pattern and maximising entropy production, we see a moist adiabatic lapse rate in the middle of the atmosphere, and a stratosphere naturally emerges. This model already gives suitable results for temperatures as the profile is similar to the IPSL-CM6A-LR model (light blue triangles) or the McClatchey (1972) standard tropical atmosphere (circles). The temperature profile remains close when adding a new constraint to compute precipitations.

## 5.3   With a constraint on precipitation (grey triangle points)

Adding the constraint on precipitation (8) is the article's novelty. Nevertheless, the philosophy remains the same: because the atmosphere is heated from below, energy fluxes (Fig. 2b) want to go upward, but they have to be opposed to the energy gradients (Fig. 2c), so these gradients want to be negative or to go to zero leading to infinite mass fluxes (Fig. 2d) in the middle of the atmosphere. However, saying that no water vapour can be created prevents infinite mass fluxes and nil energy gradients. Consequently, energy decreases (while energy fluxes are positive) below a zone we call the tropopause; above, energy increases

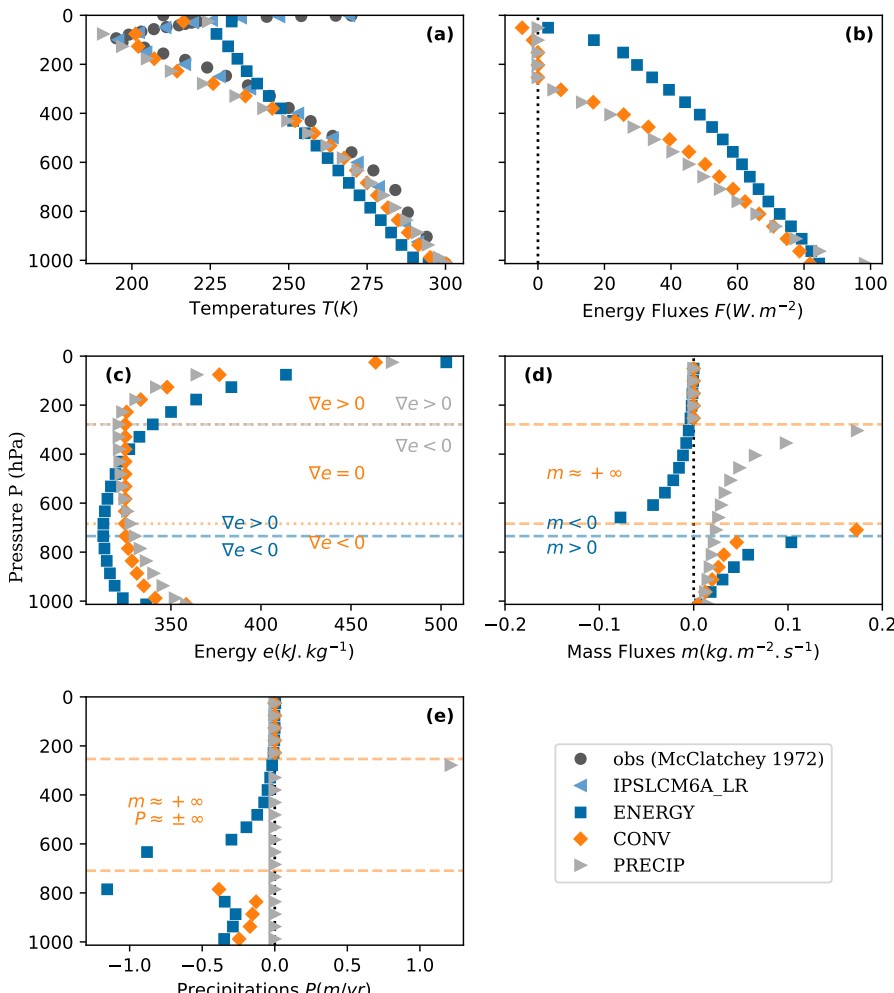

**Figure 2.** 3 different cases: 1) eq. ENERGY, 2) eq. CONV et 3) eq. PRECIP. 1) $\sigma = 53.917$, 2) $44.304$ and 3) $41.108\ \mathrm{mW.m^{-2}.K^{-1}}$.

(while energy fluxes are equal to zero). Regarding precipitation (Fig. 2e), some water vapour is taken in box number $0$ and continues going up to maximise mass fluxes. Then it reaches the tropopause, where mass fluxes become nil because there is not enough energy to go up, and then water vapour disappears, i.e., it precipitates.

The computed precipitations are equal to $1.2\ \mathrm{m.yr^{-1}}$, which is the correct order of magnitude of tropical precipitations. Comparison of this result with real-world or modelled tropical precipitation depends on the box size chosen for the tropical

area. For example, the average Earth System Model IPSL-CM6-LR precipitations between -23 and +23 degrees of latitude are $1.4\ \mathrm{m.yr^{-1}}$. For real-world data, average precipitations between 30°S and 30°N between 1980 and 1994 are $1.3\ \mathrm{m.yr^{-1}}$ (figure 8 of Xie and Arkin (1997)), and zonally averaged precipitations between 1979 and 2001 lie between 0.6 and 2 $\mathrm{m.yr^{-1}}$ (figure

5 of Adler et al. (2003)). Note that the MEP model gives less precipitation, and the other model gives more precipitation than observations. We will see that it is a common bias for models to overestimate precipitation.

Another local maximum of entropy production can give, for example, $2.2 \ \mathrm{m.yr}^{-1}$, though with less overall entropy production. This "uncertainty" in computed precipitations is probably due to the harsh resolution of only 20 boxes because choosing one box or another to precipitate leads to a different value of precipitations. To test this hypothesis, it is possible to simplify the problem by imposing the precipitation value in only one box. Then, the problem is numerically easily solved, and precipitation converges when the number of boxes increases. With 81 boxes, the computed precipitations are $1.15 \ \mathrm{m.yr}^{-1}$, and the other

local maximum of entropy production leads to precipitations of $1.21 \ \mathrm{m.yr}^{-1}$. Thus, the $1.2 \ \mathrm{m.yr}^{-1}$ precipitation is a robust value.

    To the author's knowledge, it is the first time precipitations are computed with a model using maximising entropy production and without any data-tuned parameter. At least, the fact that the good order of magnitude of precipitation can be computed with little knowledge is of prime theoretical importance for climate scientists because it means the radiative transfer, or greenhouse

gases, mainly drive atmospheric precipitations. This statement is not new and can also be deducted from Pierrehumbert (2002); Rennó et al. (1994) or eq. 9 of O'Gorman and Schneider (2008). Still, variation in surface sensible heat flux or how the surface radiative transfer depends on other processes is a source of discordance between models, leading to slightly different computed surface latent heat flux (or precipitations). Our model has less bias than radiative-convective models or the one from the atmospheric model intercomparison project, and further investigation is needed to understand if it is by chance or if there

is some profoundly hidden physics in the MEP hypothesis or the imposed constraints.

### 5.4   Doubling the $CO_2$ concentration

When looking at absolute values, the MEP approach seems to provide a good order of magnitude for climate variables. It is interesting to test if it can capture small changes in external forcing. A classic test for climate models is to look at the climate sensitivity, which is the difference of temperature at 1.50 m between conditions where $CO_2$ is at pre-industrial level (280 ppm)

and conditions where $CO_2$ is doubled (560 ppm). The only feedbacks present here are the water vapour feedback incorporated by fixing the relative humidity in the radiative code, and the lapse rate feedback. Because box number 0 is a thin surface boundary layer, we take the temperature of layer number 1, whose middle is at 988 hPa ($\sim 220$ m). The climate sensitivity is 1.1 K for eq. ENERGY, 0.7 K for eq. CONV, and 1.0 K for PRECIP (see Fig. 3). For comparison, the climate sensitivity of the state-of-the-art IPSL-CM6A-LR model is 3.0 K, which is much more. The limited sensitivity to CO2 is not surprising

since our model only represents a few amplifying phenomena. For instance, the albedo is fixed, there is no deep atmospheric convection, and the ground has no heat capacity. The ground is an infinite water reservoir, the model is 1D, and no clouds are in the atmosphere. How to integrate (some of) these additional processes in the MEP procedure remains to be investigated, as well as how this would affect climate sensitivity.

    Nevertheless, the model of Manabe and Wetherald (1967) contains physics similar to the MEP model and predicts 2.9 K

(see Table 5). The result looks better because the convective adjustment method used in Manabe and Wetherald (1967) is very efficient in transferring a rise of temperature from the top of the atmosphere into the bottom of the atmosphere, and

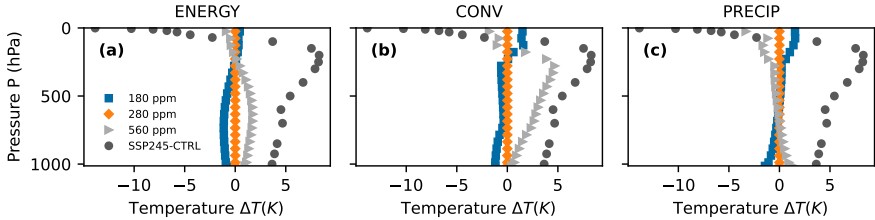

**Figure 3.** Differences of temperatures profiles between $CO_2$ at 180 ppm (or 560 ppm), and $CO_2$ at 280 ppm.

that is why the climate sensitivity, which regards only the temperature at the bottom of the atmosphere is so well predicted. In the MEP case, the temperature at the bottom of the atmosphere is much less constrained by radiative transfer, which is why the result seems less good. However, looking not at one temperature but at temperature vertical profiles, the Manabe and
320 Wetherald (1967) model computes a constant elevation of temperature in the whole troposphere (see their Figure 16), which is not at all consistent with the vertical profile computed by the state of the art IPSL-CM6A-LR model (SSP245-CTRL figure 3). Interestingly, radiative-convective models that do not use a mean moist convective adjustment (of $6.5$ K.m$^{-1}$) but a local moist adiabatic convective adjustment or a cumulus model (Lindzen et al. (1982)) predict both a better temperature profile and a considerably reduced climate sensitivity. For example, depending on the imposed parameters, Lindzen et al. (1982) find a
325 climate sensitivity of 1.4 K and 1.7 K with a moist adiabatic convective adjustment, and 1.6 K, 2.2 K, 2.3 K or 0.9 K with a cumulus model (see Table 1 and 2 of Lindzen et al. (1982)).

The vertical temperature distribution within the atmosphere is plotted for eqs. ENERGY, CONV, PRECIP figure 3, that is the difference of temperatures between a case with 560 ppm (or 180 ppm corresponding to the last glacial maximum) and a case with 280 ppm. The difference between 560 ppm and 280 ppm is also plotted for the IPSL-CM6A-LR model (SSP245-
330 CTRL), with a temperature average taken in the tropical region (between $\pm23°$). When looking at the shape of temperature distribution and comparing it to the SSP245-CTRL, the best model seems to be the problem CONV since the temperature difference increases with height and then decreases in the stratosphere. For the PRECIP model, it just decreases in the entire atmosphere. The less-constrained CONV experiment provides a better sensitivity of the temperature profile to CO2 doubling and, in that sense, does better than the PRECIP case or Manabe and Wetherald (1967) model. Results from more evoluted
radiative-convective models are similar to CONV (see, for example, figure 3 and 8 of Lindzen et al. (1982)). Considering the simplicity of the model and the absence of parameterisations, it was expected that correct values for all model outputs would not be obtained. The fact that CONV is better and compares well with similar parameterized models suggests that the PRECIP experiment is over-constraining the water cycle, something that could be relaxed by introducing new degrees of freedom. For example, the convection pattern could allow mass fluxes between layers that are not adjacent, allowing deep convection to
occur, or water vapour could be allowed to vary between zero and saturation, or the setup could be extended from 1D to 2D. We think that once this is done, the sensitivity of the temperature profile to $CO_2$ should get close to the CONV experiment. Unfortunately, this complexifies the optimisation problem, and we have yet to be able to test this idea.

Moreover, when interpreting these results, one should remember that depending on the resolution method of the variational problem (or the choice of initial conditions), results may differ by about 1 Kelvin. They differ even more by choosing an arbitrary local maximum of entropy production instead of the "supposedly" global maximum. Indeed, we note that because we changed the resolution method and found a new result with higher entropy production, the climate sensitivity of the problem CONV is a bit different than in Fig. 6 of Labarre et al. (2019). However, the purpose of this model was not to produce precise or reliable predictions. More modestly, this model aims to demonstrate that some relevant elements of the climate system (temperatures and precipitations) can be computed with only a minimal set of hypotheses, i.e. conservation laws only, without any parameter tuning.

## 5.5 Fixing the relative humidity

In the above simulations, in the energy equation 5, the specific humidity $q$ has been fixed to its value at saturation $q_s(T)$. In other words, the relative humidity is fixed to $h = q/q_s(T) = 1$. Relaxing this constraint by letting the relative humidity vary between 0 and 1 poses a numerical problem that has yet to be solved. However, one may wonder how fixing the relative humidity by hand would change the results. In this section, we solve the problem with a constraint on precipitation (PRECIP) and a relative humidity constant in the entire column but chosen between 0.1 and 1. $h = 1$ corresponds to the PRECIP setting. As before, a possible method is to test different initial conditions and see which results give the maximum entropy production. However, this method seems to fail when $h < 0.5$, and the results are not robust anymore. Fortunately, phenomenology shows that in all our simulations, the case where precipitation happens in only one box seems to always correspond to the maximum entropy production. With this knowledge, we use a simpler way to obtain the MEP solution. We use a method where the place and value of precipitations are imposed to obtain the result more directly. We find the maximum entropy production by exploring the phase space by changing the value of $P$. We use the following protocol: relative humidity $h$, precipitation $P$, and the box where precipitation occurs $n_p$ are fixed. A value of entropy production $\sigma$ is found for each set of $(h, P, n_p)$. The explored phase space is $h \in \{0.1, 0.2, ..., 1\}$, $P \in \{0.1, 0.2, ..., 10\}$, $n_p \in \{13, 14, ..., 19\}$. The number of atmospheric boxes is fixed at $n = 20$. The results are shown in Fig. 4, with the obtained values of maximum entropy production $\sigma$ and precipitations $P$. For all values of $h$, the maximum of entropy production corresponds to precipitations in box number $n_p = 15$, as in the PRECIP problem (corresponding to $h = 1$ here).

When $h$ tends to zero, precipitations seem to go to infinity. We stopped at $P = 10$ m.yr$^{-1}$, so the true value of $P$ is greater than 10 m.yr$^{-1}$ for $h \leq 0.3$. Further analysis shows that when $h$ tends to zero, the function $\sigma(P)$ has less and less a clear maximum and becomes very flat, so very different values of precipitations can give slightly different values of entropy production. This explains why results were not robust when using the classic convergence algorithm when $h$ is low: the phase space becomes very flat in the region of the maximum. The important result arising from Fig. 4 is that the maximum of entropy production corresponds to $h = 1$, and $P = 1.2$ m.yr$^{-1}$. This means that in this 1D vertical and stationary case, the system is the most efficient to create entropy when water vapour is saturated. We did not test all possible relative humidity profiles, but we expect that the MEP solution is close to $h = 1$ in each box.

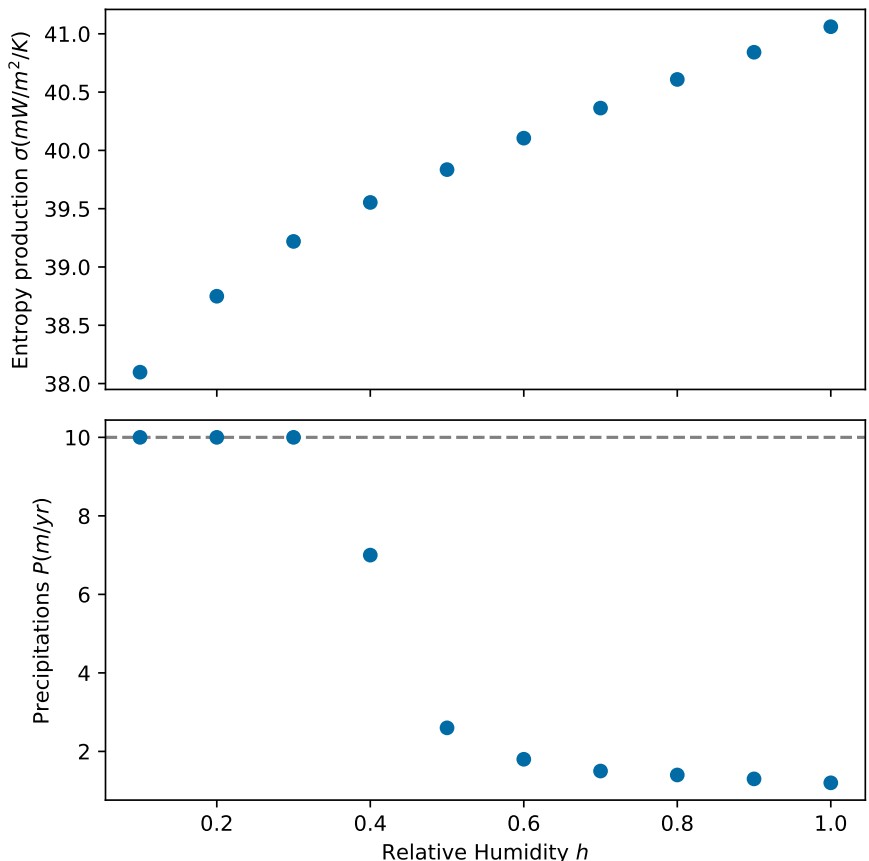

**Figure 4.** Values of maximum entropy production $\sigma$ and corresponding precipitations $P$ for each constant profile of relative humidity $h$. The maximum corresponds to precipitations in only box number 15 (with $n = 20$). The maximum between all $h$ is $h = 1$, with $\sigma = 41.108$ mW.m$^{-2}$.K$^{-1}$, and $P = 1.2$ m.yr$^{-1}$.

## 6   Discussion

The state-of-the-art GCMs and ESMs, and MEP models are based on the same conservation laws. In a GCM, the local conservation laws lead to partial derivative equations that are true in the limit of infinitely small differentials. The momentum conservation is the Navier-Stokes equation (present only to the horizontal), the energy conservation is the thermal energy equation, and the mass conservation is $\nabla \cdot u = 0$. These equations were demonstrated for infinitely small increments. However, in GCMs and ESMs, they are integrated under a grid that needs to be smaller. Indeed, because of non-linearity small scales do have an impact on large scales. Therefore, more than the first conservation equations are needed for consistent results. Additional equations involving tunable parameters are added and are sometimes called "closure equations". In the MEP framework, the energy conservation is eq. 4, and the mass conservation is immediately imposed by the convection pattern (eq. 5) and con-

straint $m \geq 0$. The water conservation is imposed by $P \geq 0$ in eq. PRECIP. So, with MEP, the conservation laws are defined as constraints of an optimization problem, and unknown variables are resolved simultaneously to reach the maximum entropy production.

Everything else in our MEP model is similar to what is done in a GCM. The radiative code is based on integrating Planck's law on different wavelength bands corresponding to different constant extinction coefficients (see supplementary materials of Herbert et al. (2013a)). The air is considered an ideal gas, and the hypothesis of hydrostatic equilibrium is made ($gdz = -\rho dp$). Of course, well-known parameters like heat capacity $C_p$ or the relation between $q_s$ and $T$ (see appendix B of Labarre et al. (2019)) are not variables of the optimization problem but just taken equal to well-established values used in GCMs and ESMs.

Still, there are many reasons why our MEP model could give different results than an ESM like the IPSL-CM6A-LR. Our MEP model has no continental surface and no energy flux between the ground and the atmosphere. Because there is no constraint on evaporation at the surface level, the ground can be seen as an infinite water reservoir, like an ocean. The ground heat flux is crucial on daily or seasonal time scales. However, it must equal zero in a stationary state without sub-surface fluxes (neglecting oceanic horizontal fluxes for ocean surfaces and geothermal fluxes on continental ones). The ground heat flux will be important if we add time to the problem to see a daily or seasonal cycle. However, this is a work beyond the scope of this article.

Also, no clouds (i.e., liquid water) are present in the air, although they are known to have an important impact on the radiative forcing. However, according to Sarachik (1978), neglecting the clouds introduces a false cooling of the surface that compensates for the false cooling by neglecting the heat transfer in the ocean. Moreover, the model is only 1D vertical and works well only for a tropical column where vertical fluxes dominate. However, O'Gorman and Schneider (2008) showed that the radiative-convective approximation is good for computing average precipitations. To refine the results, it would be easy to use fixed prescribed horizontal fluxes, but this differs from our goal. Also, there is no theoretical difficulty in extending this model to 2D or 3D and obtaining horizontal fluxes, but it is currently very technically challenging.

Finally, a reason for getting different results could be the possible lack of validity of the MEP hypothesis. That said, obtaining precipitation values close to the IPSL-CM6A-LR model or observations by Xie and Arkin (1997), 1.2 m.yr$^{-1}$ compared to 1.4 m.yr$^{-1}$ and 1.3 m.yr$^{-1}$, is very encouraging. Intuitively, given the fact that the model is built in a fully non-parametric fashion and is so minimalistic, we could have expected much more discrepancy. However, O'Gorman and Schneider (2008) showed that radiative transfer strongly constrains precipitations (see their figure 3). For example, the world's mean precipitation cannot exceed 2.1 m.yr$^{-1}$ (eq. 6 of O'Gorman and Schneider (2008)). Indeed, there is a balance at the surface between the net radiative energy budget, sensible heat flux and latent heat flux. In the tropics, the sensible heat flux is one order of magnitude less than the latent heat flux and can be neglected. The simplified energy budget allows us to give a precipitation value, given a temperature, figure 3 of O'Gorman and Schneider (2008). If the surface temperature is well constrained, which is the case in a tuned parameterised model, precipitations are also well constrained. The MEP model reproduces this simplicity: both a dominant latent heat flux and a good surface temperature. At 300 K, O'Gorman and Schneider (2008) find precipitations of about 1.8 m.yr$^{-1}$. The difference with us comes from the too-simple grey atmosphere radiative code they use. Interestingly, they prove that the same results regarding precipitations obtained from a GCM can be obtained with a radiative-convective

model (with the surface wind speed parameter chosen from the mean value of the GCM). This means the radiative-convective approach is a reasonable first approximation to investigate precipitations.

One could wonder if there is a need to add a constraint on water vapour conservation to compute precipitation. Indeed, ENERGY, CONV and PRECIP have similar surface net radiative energy budget (84, 82 and 98 $W.m^{-2}$), and saying sensible heat flux is negligible, one could already deduce the surface latent heat flux thus evaporation (i.e. precipitation). However, in

the three models, the computed sensible heat flux is not always negligible: for the ENERGY, the surface sensible heat flux is 19 $W.m^{-2}$ and the surface latent heat flux is 65 $W.m^{-2}$. For the CONV, it is 15 and 67 $W.m^{-2}$. And for the PRECIP, it is 2 and 96 $W.m^{-2}$. Evaporation is higher in the PRECIP model. Most importantly, ENERGY and CONV do not conserve water (precipitations defined as the convergence of water fluxes are either negative or infinite): evaluating the water fluxes not at the surface but at another model level would give completely arbitrary results.

The MEP model could be improved by exploring several approaches. First, the specific humidity of water vapour $q$ could be chosen not equal to saturation. Then, it needs to be clarified if an additional constraint on precipitation should be imposed, for example, saying that precipitation can occur only if $q = q_s$, that is $(q - q_s)P = 0$, where $P$ are precipitation. Moreover, this constraint is highly non-convex and numerically very harsh to solve. Second, convection is not allowed between every layer because air masses are compelled to move to adjacent layers. However, in the tropics, there is a phenomenon called deep

convection, where an air mass can go adiabatically from the bottom to the top of the troposphere. This phenomenon is not authorized with the convection pattern imposed here. However, this could be implemented by changing equation 5 and adding fluxes for non-adjacent layers.

## 7    Conclusions

Since Ozawa and Ohmura (1997), many improvements have been made. Taking a more realistic radiative code (Herbert et al.,

2013a) leads to a stable atmosphere: the potential temperature increases with altitude. Adding a convection pattern (Labarre et al., 2019) gives much more realistic results in terms of temperature and reveals a stratosphere up to $\approx 250 \ hPa$ where no convection occurs. Then, imposing a constraint on water vapour conservation leads to precipitation as close as or closer to observations than a GCM or an ESM would find. These results look great in absolute value, but results seem less satisfying when performing a sensitivity experiment such as looking at the temperature difference when doubling $CO_2$. We hope future

work adding degrees of freedom in the model should solve this issue. Nonetheless, this article demonstrates for the first time that it is possible to compute tropical precipitations in a radiative-convective model from conservation laws only, replacing the usual atmospheric parameterisations with an optimisation procedure (MEP hypothesis).

Several approaches, such as changing the convection pattern and letting relative humidity vary, need to be explored in the future. Along these lines, it might be possible to build a climate model 2D or 3D, with a representation of clouds, vegetation,

and oceans, whose "closure equations" would not at all be based on "tuned towards observations" parameters. The way clouds could be added without using any parameter is complicated. However, letting the relative humidity vary between zero and saturation, adding fluxes between non-adjacent boxes (i.e. deep convection) or making the problem 2D or 3D is mathematically

straightforward. However, in such more complex settings, we have to solve a non-linear and non-convex optimisation problem with many more variables. Currently, technical difficulties appear in the algorithm's convergence: the less convex the problem is, the more variables and non-linearities there are, the harder it is to find the absolute global maximum. This technical problem might be solved using more specific optimisation algorithms or finding a more straightforward mathematical but equivalent formulation.

Such a model could be used in contexts where tuning is impossible, such as other planets or paleoclimates. The only mandatory knowledge is the atmosphere's chemical composition, the value of the solar radiation and the ground albedo. This simplicity opens new and exciting perspectives in paleoclimate or exoplanet climate modelling.

*Code availability.* The code used to produce the results can be found at https://doi.org/10.5281/zenodo.7995540

## Appendix A: A global and a local maximum

*Author contributions.* Didier Paillard is the brain having the global understanding. Karine Watrin implemented many possibilities in the resolution code. Quentin Pikeroen also worked on the code, used it to get the article's results, and wrote the manuscript.

*Competing interests.* The contact author has declared that none of the authors has any competing interests.

*Acknowledgements.* We thank Bérengère Dubrulle for insightful discussions.

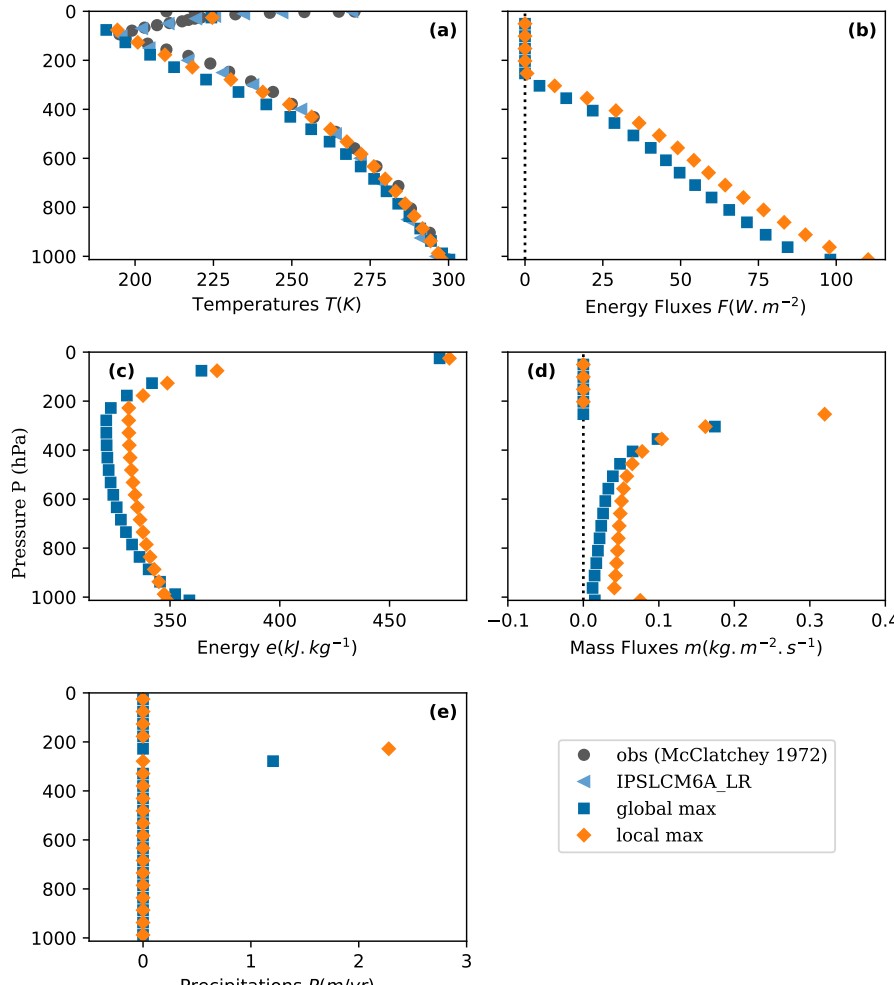

**Figure A1.** 2 different cases: 1) a global maximum of entropy production $\sigma = 41.108$ mW.m$^{-2}$.K$^{-1}$. 2) A local maximum of entropy production $\sigma = 40.078$ mW.m$^{-2}$.K$^{-1}$.

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
