# Peer review of "A radiative-convective model computing precipitations with the maximum entropy production hypothesis."

_EGUsphere, 2023_

## Author Response (AR1)

**Response to Reviewer 1**

**General comment**

The comments raised by the three reviewers share some common generic features, particularly concerning our model's very simplified setting and the applicability of its results. Thus, it appears that the manuscript's main goal was not clear enough and that the paper needs a better introduction. Before answering more specific comments made by the reviewers, we are first addressing this common concern.

The traditional way to compute the state of the atmosphere is to use physical laws (mainly conservation laws) and more empirical ones (parameterisation of fluxes). In our case, this concerns energy and water fluxes. At the surface, models use so-called bulk formulas involving « turbulent » or « drag » coefficients to compute evaporation (or latent heat) and sensible heat separately. Those coefficients are tuned on observations. In the atmosphere, these fluxes are usually calculated from gradients using a diffusion law, with a diffusion coefficient adjusted on observations. Since the physical (conservation) laws are the same for all climate models, the main differences between them are mostly linked to the tuning of these parameters: differences are therefore linked to different model settings, different choices of parametric formula, different tuning strategies, different tuning data sets, etc. It is probably one of the main problems in climate modelling.

The main advantage (and difficulty) of MEP is that we do not need a formula to directly compute these fluxes like evaporation or sensible heat. In contrast, they are obtained from a global optimisation procedure. The fact that this procedure produces reasonable energy fluxes has already been shown in previous publications, as explained in the manuscript. The fact that this procedure can also produce suitable water fluxes has never been shown before. This point is the key point of our manuscript, and we obviously need to emphasise it better and explain its novelty in a revised manuscript.

**Specific answers**

*This article presents a 1D radiative-convective model based on the thermodynamic hypothesis of maximum entropy production. I was pleased to receive the review assignment, but found it challenging. The text was disjointed and unclear, with numerous unfamiliar citations in the earlier sections that required further investigation. After several attempts, I managed to complete my reading. In my understanding, this model serves as a tool for simulating the earth system from scratch. This innovative work holds potential significance for scientific exploration; however, its practical applicability is questionable.*

We are deeply sorry that the text was hard to follow. This approach is indeed unusual in the model development community, and the manuscript was probably not explicit enough. We will clarify the context and the novelties in the upcoming version. Indeed, this model is built to serve as a tool for simulating the earth system from scratch; in this manuscript, we start with only a vertical column to construct a radiative-convective atmospheric model based on maximum entropy production (MEP). The model currently does not aim at being directly « applicable », but it is a critical step towards a new way to compute the atmospheric state.

*1 - The article suggests that energy conservation and mass conservation have been confirmed in previous studies while introducing a new constraint on the water cycle. Are these two conservation results validated by prior publications?*

Yes, indeed. Energy conservation and mass conservation have been successfully used in previous studies published by Herbert et al. (2013) and Labarre et al. (2019) to compute the atmospheric vertical temperature profile in a tropical context. Our manuscript further develops this methodology with the addition of water conservation. Interestingly, this naturally leads to a reasonable estimate of precipitations.

*2 - The updated model is tested in tropical regions. It is claimed that the results align with what happens in the tropics – does this refer to average conditions or specific locations within this region?*

Because the model is only 1D vertical, we have tested it in the tropical area, where vertical processes are predominant, and horizontal fluxes can be neglected in a first approximation. It is a classical assumption for radiative-convective models. For consistency, we used the same model setting as in the two previous studies, Herbert et al. (2013) and Labarre et al. (2019), which is a « standard » tropical atmosphere. In particular, standard $CO_2$, $O_3$ and relative humidity profiles are used for the radiative computation, and a standard temperature profile is used to compare with observations. These « standard atmospheric profiles » (McClatchey 1972) are, of course, based on averaging observations in the tropical band.

*3 - Can it be applied universally across all tropical areas?*

Suppose albedo, $CO_2$, $O_3$ and humidity profiles are given at a specific location, assuming that horizontal fluxes are negligible. In that case, we expect our model to work in the same fashion as in this « standard » setting. Nevertheless, the maximum entropy production (MEP) hypothesis can likely only be applied in a stationary setting; therefore, the results should be interpreted only in a climatological context, not in a meteorological one. It will be made more explicit in a revised version of the manuscript.

*4 - Furthermore, authors propose that "it could be used for climates where few are known, such as paleoclimate or climates of other planets". I am skeptical about whether successful testing in Earth's tropics can be directly applicable to another planet?*

This sentence was misleading and needs clarification. What we meant by this sentence was that, in contrast to other radiative-convective models, our model does not require any tuning of parameters (diffusivity coefficients, drag coefficients, boundary layer parameterisation, etc.) and, therefore, can be applied in contexts where such tuning is impossible, for example, paleoclimates or other planets. Of course, it does not mean it is directly applicable in such contexts. For instance, for other planets, the radiative code would have to be adapted to the atmosphere's radiatively active chemicals, which would not be the same as on Earth. The correct albedo should be prescribed, and many other things would likely need to be adapted.

Most importantly, the atmosphere should be sufficiently « turbulent » for the maximum entropy production hypothesis to be applicable since it corresponds in some way to maximising mixing under constraints. The fact that this non-parametric method provides reasonably good results for the Earth's present atmosphere is encouraging. It suggests it could also work for very different Earth's climatic regimes or on other planets, though this would require further specific investigations.

*5 - Regarding its limited sensitivity to CO2 increase, this outcome is expected due to insufficient consideration of physical processes within the model.*

This point was also mentioned by reviewer #2 (item #2), therefore our identical response :

The limited sensitivity to CO2 is not surprising since our model only represents a few amplifying phenomena. For instance, the albedo is fixed, there is no deep atmospheric convection, and the ground has no heat capacity. It is an infinite water reservoir, the model is 1D, and no clouds are in the atmosphere. How to integrate (some of) these additional processes in the MEP procedure remains to be investigated, as well as how this would affect climate sensitivity. Considering the model's simplicity, its deficit in reproducing the climate sensitivity may have many reasons. But we hope that in a 2D setting or with the addition of deep convection, this problem can be solved naturally. Indeed, as explained in the manuscript (lines 259-260), the less-constrained CONV experiment provides a better sensitivity of the temperature profile to CO2 doubling. This suggests that the PRECIP experiment is over-constraining the water cycle, something that would be relaxed when introducing new degrees of freedom, like (for instance) deep convection. Unfortunately, this complexifies the optimisation problem, and we have yet to be able to test this idea.

However, the purpose of this model was not to produce precise or reliable predictions. More modestly, this model aims to demonstrate that some relevant elements of the climate system (temperatures and precipitations) can be computed with only a minimal set of hypotheses, i.e. conservation laws only, without any parameter tuning.

*Once again, I question the utility of this model given these abovementioned concerns. Based on these considerations, I regretfully cannot recommend accepting this manuscript at present.*

This manuscript demonstrates for the first time that it is possible to compute tropical precipitations in a radiative-convective model from conservation laws only, by replacing the usual atmospheric parameterisation with an optimisation procedure (MEP hypothesis). We believe that this result opens new and interesting perspectives in climate modeling, in particular the possibility of building simplified atmospheric models without any tuning parameters, something that could be extremely useful in many different contexts.

References :

Herbert, C., Paillard, D., and Dubrulle, B.: Vertical Temperature Profiles at Maximum Entropy Production with a Net Exchange Radiative350Formulation, Journal of Climate, 26, 8545–8555, https://doi.org/10.1175/JCLI-D-13-00060.1, 2013b.

Labarre, V., Paillard, D., and Dubrulle, B.: A radiative-convective model based on constrained maximum entropy production, Earth SystemDynamics, 10, 365–378, https://doi.org/10.5194/esd-10-365-2019, publisher: Copernicus GmbH, 2019

**Response to Reviewer 2**

**General comments**

The comments raised by the three reviewers share some common generic features, particularly concerning our model's very simplified setting and the applicability of its results. Thus, it appears that the manuscript's main goal was not clear enough and that the paper needs a better introduction. Before answering more specific comments made by the reviewers, we are first addressing this common concern.

The traditional way to compute the state of the atmosphere is to use physical laws (mainly conservation laws) and more empirical ones (parameterisation of fluxes). In our case, this concerns energy and water fluxes. At the surface, models use so-called bulk formulas involving « turbulent » or « drag » coefficients to compute evaporation (or latent heat) and sensible heat separately. Those coefficients are tuned on observations. In the atmosphere, these fluxes are usually calculated from gradients using a diffusion law, with a diffusion coefficient adjusted on observations. Since the physical (conservation) laws are the same for all climate models, the main differences between them are mostly linked to the tuning of these parameters: differences are therefore linked to different model settings, different choices of parametric formula, different tuning strategies, different tuning data sets, etc. It is probably one of the main problems in climate modelling.

The main advantage (and difficulty) of MEP is that we do not need a formula to directly compute these fluxes like evaporation or sensible heat. In contrast, they are obtained from a global optimisation procedure. The fact that this procedure produces reasonable energy fluxes has already been shown in previous publications, as explained in the manuscript. The fact that this procedure can also produce suitable water fluxes has never been shown before. This point is the key point of our manuscript, and we obviously need to emphasise it better and explain its novelty in a revised manuscript.

**Specific answers**

*General*

*The authors have extended a simple 1-d climate model to include precipitation. The model is based on a relatively complex radiation scheme in combination with an optimization of entropy production as a closure representing all other processes defining the temperature profile. The precipitation results from an additional constraint imposed to the optimization. The approach follows previous work utilising the same model and relays on the maximum entropy production (MEP) conjecture.*

*Although not proven to work for the climate system, MEP has been successfully applied to various problems. Previous work with the same model (but without the precipitation constraint) seem to demonstrate the applicability of this approach (Herbert et al. 2013 & Labarre et al. 2019, as cited by the authors). Thus, in general, the idea followed by the authors seems well founded. However, I have to admit that to me the results presented are not significant enough to warrant publication in the present form. So far, it is hard for me to extract the gain of new knowledge provided by this study/model. I have detailed some points below.*

We thank the reviewer for these detailed and useful comments.

*Major*

*1) Results: It appears that most of the results described in sections 5.1 and 5.2 have been already (and more thoroughly) discussed in Herbert et al. (2013) and Labarre et al. (1019). The authors need to make clearer what is new here.*

Yes, the results in sections 5.1 and 5.2 have already been obtained by Herbert et al. (2013) and by Labarre et al. (1019). Our results (section 5.3 and following) are a further development based on these previous publications, and we wanted to highlight the progress obtained when putting successive additional physical constraints in this model set. As explained above, these previous papers successfully used the MEP hypothesis to compute energy fluxes. The main novelty of our manuscript is to demonstrate that the same MEP procedure can also be used to calculate water fluxes successfully, something that has never been done before. In a new version of the article, we will clarify previous contributions and the novelty of this manuscript.

*2) Section 5.4, sensitivity: In my view, the results of the sensitivity study (2xCO2) question the approach used by the authors (either MEP in general or the imposed constraints). I appreciate the honesty in showing these results. However, I think the deficit of the model in reproducing the climate sensitivity and, in particular, the temperature profile is too serious to postpone the investigation of the reasons (and a potential fix) to further investigations (as the authors do e.g. in L251). From the results I would simply conclude that the method as it is now does not work sufficiently well.*

It is not clear what « sufficient well » means. Conserving water (experiment PRECIP) is likely « much better » than not doing so (experiment CONV or ENERGY). Computing reasonable precipitations is also probably « much better » than not doing so. Our results are, therefore, a significant positive increment with respect to previously published similar models. But, this does not prevent us from evaluating other aspects of the model.

But the point of climate sensitivity was also mentioned by reviewer #1 (item #5); therefore, our identical response is below:

The limited sensitivity to CO2 is not surprising since our model only represents a few amplifying phenomena. For instance, the albedo is fixed, there is no deep atmospheric convection, and the ground has no heat capacity. It is an infinite water reservoir, the model is 1D, and no clouds are in the atmosphere. How to integrate (some of) these additional processes in the MEP procedure remains to be investigated, as well as how this would affect climate sensitivity. Considering the model's simplicity, its deficit in reproducing the climate sensitivity may have many reasons. But we hope that in a 2D setting or with the addition of deep convection, this problem can be solved naturally. Indeed, as explained in the manuscript (lines 259-260), the less-constrained CONV experiment provides a better sensitivity of the temperature profile to CO2 doubling. This suggests that the PRECIP experiment is over-constraining the water cycle, something that would be relaxed when introducing new degrees of freedom, like (for instance) deep convection. Unfortunately, this complexifies the optimisation problem, and we have yet to be able to test this idea.

However, the purpose of this model was not to produce precise or reliable predictions. More modestly, this model aims to demonstrate that some relevant elements of the climate system (temperatures and precipitations) can be computed with only a minimal set of hypotheses, i.e. conservation laws only, without any parameter tuning.

*3) Discussion and conclusions: I do not see sufficient information on significant new results in the discussion and conclusions. From the discussion, I mainly learn that there can be many (more or less obvious) reasons why the method may not work (and the key result, P of reasonable order, is a "surprise" (L288)). The conclusions appear to question the key result (L310-311) and give a quite general outlook only. Overall, this is a bit disappointing.*

Considering that MEP has never been used before to compute precipitations and that we put so little physics in the model, the fact that the calculated precipitations are close to observations is indeed an excellent surprise. In this sense, we conclude that the method « works extremely well » compared to the little input of the model.

But as noted by the reviewer, the method « may not work » for all aspects of climate, and we are certainly not claiming so. More generally, we are not claiming that the MEP hypothesis is a magic bullet to simulate climate. Our manuscript attempts to evaluate its relevance to climate science by building « minimal » models in a fully non-parametric fashion. The fact that a « minimal model » may not work on all aspects is quite natural and expected. The fact that a « minimal model » works quite well on some parts is much more surprising.

*Minor*

*1) The abstract needs to be shorten significantly. Most of it might move to the introduction.*

The abstract will be shortened in the updated article version.

*2) A figure like Fig. 1 in Labarre et al. (2019) may help the reader to understand the setup of the model.*

It is a helpful remark, and a revised article version will include a new figure like that of Labarre et al. (2019).

*3) From Fig 1b) it seems that all models show about the same energy flux at the surface, which, in PRECIP, seems to be approx. the latent heat flux consistent with the evaporation/precipitation (as one may expect). That means, all models would predict approx. a similar P by using this quantity and I am wondering whether this would be a fairer comparison instead of using a quantity (water vapour flux convergence) which is not a part of ENERGY & CONV.*

The total energy flux at the surface varies slightly between models. However, the decomposition between sensible and latent heat flux is sometimes different.

ENERGY : sensible heat flux: 19 W/m2 & latent heat flux: 65 W/m2 (0.82 m/yr)

CONV: sensible heat flux: 15 W/m2 & latent heat flux: 67 W/m2 (0.85 m/yr)

PRECIP: sensible heat flux: 2 W/m2 & latent heat flux: 96 W/m2 (1.2 m/yr)

Evaluating precipitations from surface evaporation could be possible from ENERGY and CONV but would give rather different results. Most importantly, ENERGY and CONV do not conserve water (precipitations defined as the convergence of water fluxes are either negative or infinite): evaluating the water fluxes not at the surface but at another model level would give completely arbitrary results. That is why we developed the PRECIP model setting.

The model predicts, therefore, consistent precipitations. It also indicates that they should form at the top of the troposphere in our model configuration.

In a revised version, we will emphasise these points.

*4) Fig 1b: adding a moist adiabat may be helpfull. In addition: Do the lower most values represent the surface temperatures (i=0)? If not they may be added.*

In Figure 1c, the moist adiabat corresponds to a constant energy (a vertical line). The CONV experiment (orange diamonds) follows such a moist adiabat in the free troposphere.

And yes, for temperature and energy, the lower value (i=0) corresponds to the surface.

*5) L239/240: I think the fact that the surface radiation budget controls the evaporation (and thus the water cycle) is well known. I do not completely understand why this needs to be emphasised as "of prime theoretical importance"*

Indeed, the surface radiation budget controls evaporation. However, there is no obvious way to differentiate how much is transferred into sensible heat or into latent heat. As explained in the first paragraph, using a parameterised formula is the traditional way to compute those fluxes. At the surface, models use a so-called bulk formula involving « turbulent » or « drag » coefficients to calculate evaporation (latent heat) and sensible heat separately. Those coefficients are tuned on observations.

In contrast, with the MEP hypothesis, fluxes are obtained from a global optimisation procedure. This was already tested for energy fluxes but not for water fluxes. The fact that this MEP procedure can also produce reasonable water fluxes has never been shown before and is indeed of prime theoretical importance. This is the key point of our manuscript; we obviously need to explain this better in a revised manuscript.

*6) I might be wrong but it seems for me that it might be possible to use a q not equal to qs (e.g. q=rh\*qs, rh=const.) also in the precipitation case (and still have P=FW(i+1)-FW(i)). If so, I'm wondering about the sensitivity of the results to rh.*

Yes, it is indeed possible to relax the assumption that relative humidity is lower than one, and we could discuss this point in a revised article. Reducing relative humidity rh tends to decrease entropy production. In our model setting, the MEP state corresponds to rh equal to, or close to one everywhere in the troposphere.

*7) L307: "[...] leads to a stable atmosphere: the potential temperature decrease with altitude." I guess a typo, as stable means pot. temp. increases with z.*

Yes, it is a typo.

References :

Herbert, C., Paillard, D., and Dubrulle, B.: Vertical Temperature Profiles at Maximum Entropy Production with a Net Exchange Radiative350Formulation, Journal of Climate, 26, 8545–8555, https://doi.org/10.1175/JCLI-D-13-00060.1, 2013b.

Labarre, V., Paillard, D., and Dubrulle, B.: A radiative-convective model based on constrained maximum entropy production, Earth SystemDynamics, 10, 365–378, https://doi.org/10.5194/esd-10-365-2019, publisher: Copernicus GmbH, 2019

**Response to Reviewer 3**

**General comments**

The comments raised by the three reviewers share some common generic features, particularly concerning our model's very simplified setting and the applicability of its results. Thus, it appears that the manuscript's main goal was not clear enough and that the paper needs a better introduction. Before answering more specific comments made by the reviewers, we are first addressing this common concern.

The traditional way to compute the state of the atmosphere is to use physical laws (mainly conservation laws) and more empirical ones (parameterisation of fluxes). In our case, this concerns energy and water fluxes. At the surface, models use so-called bulk formulas involving « turbulent » or « drag » coefficients to compute evaporation (or latent heat) and sensible heat separately. Those coefficients are tuned on observations. In the atmosphere, these fluxes are usually calculated from gradients using a diffusion law, with a diffusion coefficient adjusted on observations. Since the physical (conservation) laws are the same for all climate models, the main differences between them are mostly linked to the tuning of these parameters: differences are therefore linked to different model settings, different choices of parametric formula, different tuning strategies, different tuning data sets, etc. It is probably one of the main problems in climate modelling.

The main advantage (and difficulty) of MEP is that we do not need a formula to directly compute these fluxes like evaporation or sensible heat. In contrast, they are obtained from a global optimisation procedure. The fact that this procedure produces reasonable energy fluxes has already been shown in previous publications, as explained in the manuscript. The fact that this procedure can also produce suitable water fluxes has never been shown before. This point is the key point of our manuscript, and we obviously need to emphasise it better and explain its novelty in a revised manuscript.

**Specific answers**

*General comments*

*Pikeroen et. al. proposed a radiative-convective model with no tunable parameter needed based on Maximum Entropy Production (MEP) theory. MEP is an excellent theory, and it has been successfully demonstrated to simulate different physical processes in previous studies. Previous applications of MEP theory demonstrated that the number of parameters can be significantly reduced because the information is efficiently used. Therefore, MEP-based model can potentially reduce the parametric uncertainty in Earth system model simulations/projections, which stem from hundreds of uncertain parameters.*

We certainly agree with the reviewer on these points.

*However, the derivation of the MEP model in this study involves a lot of assumptions, constraining the application of the proposed model in an atmospheric model to simulate realistic process.*

For the first time, we aimed to demonstrate the possibility of computing precipitations using MEP as a closure hypothesis. It requires a « minimal model » of the atmosphere, therefore the very

simplified setting we used in this study: a (rather classical) radiative-convective model, enforcing conservation laws, but with no turbulent parameters in contrast to usual practice.

*Although the authors mentioned the limitations in the discussion section, I cannot see the potential solution to resolve the limitation to extend the proposed model to 2D domain with varying humidity profile and surface conditions. This study will be a significant contribution to atmospheric modeling if they can reduce their assumptions and demonstrate the application in more realistic conditions. Otherwise, I cannot see the motivation of developing the MEP-based radiative-convective model. Overall, I cannot recommend the publication of this study in current form.*

The simplifying assumptions used to derive our MEP model have been chosen to make it as simple as possible and to enable us to compute and interpret the outcome easily. Indeed, this comes at the cost of less realism. But, there is no theoretical limitation to go much further. For example, letting the relative humidity vary between zero and saturation, adding fluxes between non-adjacent boxes (i.e. deep convection) or making the problem 2D or 3D is mathematically straightforward: in such more complex settings, we still have to solve a (non-linear and non-convex) optimisation problem with (many) more variables. But technical difficulties then appear in the convergence of the black-box algorithm we are currently using (scipy. optimize): the less convex the problem is, the more variables and non-linearities there are, the harder it is to find the absolute global maximum. This technical problem could probably be solved using more specific optimisation algorithms, but finding or developing such numerical methods is beyond the scope of our study. Since this is the first time anyone has explored this research direction, starting with simple model settings and standard numerical algorithms seems natural.

*Specific comments*

*The model is very simplified with many processes ignored, and the assumptions will not be valid for realistic application. For example, Line 97 assumes energy budget is a function of the temperature, with relative humidity and surface albedo being constant. However, energy budget is significantly affected by the surface conditions and the relative humidity is not only affected by temperature.*

We fully agree with the reviewer that reality is more complex than our model configuration. Indeed, the temperature only affects humidity since relative humidity is fixed to one (saturation). In fact, we can relax this saturation assumption, but this does not change our results in our current model setting (see response to reviewer #2, item #6). To obtain more realistic results, including deep convection (in 1D) and/or horizontal fluxes (in 2D or 3D) would be necessary. This will be discussed in a revised version of the paper.

*I cannot understand why and should be taken as zero. Specifically, the shortwave radiation is from the sun, which represents positive flux from space to atmosphere. In addition, there are non-zero energy flux transferred from surface to subsurface of the Earth, i.e., ground heat flux. One may argue the ground heat flux are averaged to be zero in long-term, but it is not always zero in land surface energy balance. It will be helpful for the authors to clarify their assumptions.*

The energy budget is the sum of incoming and outgoing radiative fluxes. The ground heat flux is critically important on daily or seasonal time scales. However, it must equal zero in a stationary state without sub-surface fluxes (neglecting oceanic horizontal fluxes for ocean surfaces and geothermal fluxes on continental ones). The maximum entropy production (MEP) hypothesis can be applied more efficiently in a stationary setting. Therefore, our choice is to consider only the average state of the system. The ground heat flux will be important if we add time to the problem to see a

daily or seasonal cycle. But this is a work beyond the scope of this article. Again, this would be clarified in a revised version.

*Line 135 assumes specific humidity to be the saturated specific humidity. This assumption is not consistent with Line 96. Or is used at Line 96? Overall, I don't think it is reasonable to assume the air is saturated when computing the specific energy.*

Indeed, the relative humidity is taken at saturation in the MEP problem (line 135), but it is fixed at a standard value in the radiative code (line 96). This decoupling may be inconsistent, but it is common in the climate community or General Circulation Models to examine models in a decoupled fashion before studying fully coupled model configurations. We choose to proceed also in this way to keep the problem simpler and to be able to interpret results more easily.

*Line 163 assumes the evaporation of the layer balances the total precipitation from layer . This assumption ignores the vapor movement in horizontal direction. Therefore, the proposed model may only work for very large scales that horizontal vapor movement is negligible than the evaporation for the precipitation. If so, the author should clarify this limitation.*

Indeed, this is a 1D model without prescribed horizontal fluxes. It would be easy to use fixed prescribed horizontal fluxes to represent some specific location or situation, but this differed from our goal. As explained above, there is no theoretical difficulty in extending this model to 2D or 3D and obtaining horizontal fluxes, but currently, this is very challenging technically. Therefore, our model is 1D, and as noted by the reviewer, it can only be compared with situations where horizontal fluxes are much smaller than vertical fluxes. That is why we restrict our discussion to the averaged tropical area.

*Line 248 – Line 249: could this substantial underestimation caused by the assumptions of the model, for example, fixed relative humidity profile and albedo? I think the climate sensitivity is a very important metric to evaluate a model. If the climate sensitivity cannot be well captured due to the model assumptions, then those assumptions are not valid.*

Considering the simplicity of the model and the absence of parameterisations, it was expected that correct values for all model outputs would not be obtained, as shown in the manuscript with climate sensitivity. We aimed to provide a fair evaluation of the MEP closure hypothesis when applied to the water cycle in our simple model. We could have chosen to present only positive results and not negative ones, but it seems more honest to discuss also the limitations of the current model setting. As explained above, results should likely get closer to expectations once the model is complexified.

*Line 254: where is the plot for the IPSL-CM6A-LR?*

In Figure 2, the plot of the IPSL-CM6A-LR model results are shown as dark circles with the caption SSP245-CTRL. We will make this more evident in a revised version.